# From Access Control to Usage Control with User-Managed Access

### Access Control on the Web

Solid leverages Semantic Web technologies to enable interoperable storage, access and (re)use of resources within a decentralized global ecosystem [1]. Originating from before the dawn of modern access control —when Authorization header schemes were the most elaborate mechanism available for HTTP— the project came up with WAC [2] and ACP [3], languages for writing policy lists, with an algorithm for evaluating those. Around the same time —in the early 2000s— enough developers started bumping into the limits the HTTP Authorization header (as well as initial proprietary lock-in 'solutions') to spark the creation of OAuth [4], today's de facto standard for access control on the Web. However, just like authorization was an explicit non-goal in HTTP, OAuth did not provide authentication as a service. This was later remedied by the OpenID initiative, which constructed an identity layer on top of it in the form of OpenID Connect [5].

While the Solid project incorporated aspects of both OAuth and OIDC in their specifications, it has missed some key elements that make those standards so widely adopted. Both WAC and ACP lack a separation of concerns between resource servers and authorization servers, leading to a request-efficient but inflexible system. By synchronously evaluating access controls based on a resource request and accompanying authentication token over the policy documents stored in the resource hierarchy of the Solid pod, the ability to request and evaluate claims —outside webid, identity provider and client provided by the authentication token— is restricted. Access control management is tailored to the specific interface(s) (protection domain) of a single resource server, and therefore impractical to manage and audit over multiple servers. The choice of policy language, evaluation algorithm, and authentication options are tightly bound to the evolution of that specific resource server, to which authentication details are unnecessarily revealed. Moreover, they rely on a hierarchical resource structure, which assumes a read-write symmetry, and prevents full independence of data and application, leading to a proliferation of non-interoperable application-specific APIs on top of Solid.

### Moving towards Usage Control

Moving towards data sharing ecosystems that enable users to define requirements for both access and usage control to manage their resources on the Web requires building further the current standards. OAuth 2.0 provides a robust framework for managing delegated access grants, but misses the identity features required to build data sharing ecosystems for the Web. OpenID

*3nd Solid Symposium, 24 − 25 April 2025, Leiden*

Connect (OIDC) extends these capabilities through inclusion of user identity, but restricts its claim management to fixed set of identity claims as defined in the specification the issuing of which is restricted to the OIDC Identity Provider. Enabling users to define requirements for access and usage control over their data on the Web requires improving the flexibility offered to request and provide claims, and benifits greatly from the flexibility to integrate alternative standards such as Decentralized Identifiers (DIDs) [6] and Verifiable Credentials (VCs) [7].

The User-Managed Access [8] specification extends Oauth 2.0. Due to its flexibility, UMA address limitations such as identity lock-in and synchronous access delegation. It is there possible to have asynchronous access delegation to third parties, allowing resource owners to predefine access control policies. Through interactive claims gathering, UMA allows requesting parties to present a variety of claims, unrestricted by a single identity provider. Furthermore, by separating a user into the roles of requesting party and resource owner, UMA allows facilitates access to other users, rather than only applications. Additionally, as UMA does not define the policy assessment, it is essential for the AS to adopt capable of integrating those generic claims effectively. As a W3C Recommendation, The Open Digital Rights Language (ODRL) [9] is prime candidate. Not only is it possible to combine claims within the policy, it furthermore allows expressing usage control rules rather than just access control. This allows to enforce the full lifecycle of data use; such as the purpose of access and fine-grained constraints including the time dimension.

### Integrating Usage Control within Solid

Within the Solid Protocol, the Solid-OIDC specification [10] already advocates for the adoption of User-Managed Access. We implemented an open-source UMA prototype governing usage control to Solid servers through integration with the Community Solid Server (CSS) [11]: https://github.com/SolidLabResearch/user-managed-access/. Internally, the Authorization Server is comprised of three parts: (i) The **Credential Verifier** that can verify any kind of claims (including Verifiable Presentations), (ii) the **Policy Engine** that evaluates ODRL policies using the ODRL Evaluator [12] and (iii) the **Negotiator Component**, which governs the interaction with the client, towards a mutually beneficial way agreement.

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
