# OpenReview forum: "From Access Control to Usage Control with User-Managed Access"
_SolidProject.org/SoSy/2025/Privacy_Session — SoSy2025-Privacy_

### Official Review · ~Ross_Horne1 · 2025-03-17
**UMA for Solid**

**Rating:** 9
**Confidence:** 5

**Review:**

The paper combines UMA and ODRL to address limitation with the existing Solid OIDC and WAC/ACP.

It's well know that the current Solid authentication standard have limitations. The combination of technologies proposed seems to be natural.

My suggestion is that the differences between Solid OIDC and UMA+ODRL should be laid out more systematically in the table. Some of these differences can be justified precisely with respect to threats countered by choices. However, since this is just an abstract for a talk the submission is sufficient.

---

### Official Review · ~Anelia_Kurteva1 · 2025-03-19
**Towards usage controls in SOLID**

**Rating:** 8
**Confidence:** 4

**Review:**

The abstract provoked my interest in the topic and I would be interested in hearing more about the it at the session.

Minor comments:
The presentation of the content could have been more accessible to a general audience, as the title doesn't clearly indicate the focus on SOLID or decentralised technology.

The Github of the UMA is well documented-there are guidelines on the steps that need to be followed to test the implementation and a demo as well, which is a plus.

Having a comparative table of different approaches as suggested by the other reviewer would be beneficial.

Since this is an extended abstract for a talk, I think the authors should include one or two sentences of what exactly they will present (e.g. similar to the content of abstracts in papers) - a short summary with their objective.

Terms should be defined at the start and used consistently in the paper.

---

### Decision · Program_Chairs · 2025-04-01

Accept